# Impact of Nutrition on Short-Term Exercise-Induced Sirtuin Regulation: Vegans Differ from Omnivores and Lacto-Ovo Vegetarians

**DOI:** 10.3390/nu12041004

**Published:** 2020-04-05

**Authors:** Arne Björn Potthast, Josefine Nebl, Paulina Wasserfurth, Sven Haufe, Julian Eigendorf, Andreas Hahn, Anibh Das

**Affiliations:** 1Clinic for Paediatric Kidney, Liver, and Metabolic Diseases, Hannover Medical School, 30625 Hannover, Germany; Potthast.Arne@mh-hannover.de; 2Institute of Food Science and Human Nutrition, Leibniz University Hannover, 30167 Hannover, Germany; nebl@nutrition.uni-hannover.de (J.N.); wasserfurth@nutrition.uni-hannover.de (P.W.); hahn@nutrition.uni-hannover.de (A.H.); 3Institute of Sports Medicine, Hannover Medical School, 30625 Hannover, Germany; Haufe.Sven@mh-hannover.de (S.H.); Eigendorf.Julian@mh-hannover.de (J.E.)

**Keywords:** sirtuins, vegetarian, vegan, exercise, endurance athletes, metabolic regulation

## Abstract

Both nutrition and exercise are known to affect metabolic regulation in humans. Sirtuins are essential regulators of cellular energy metabolism; SIRT1, SIRT3, and SIRT4 have a direct effect on glycolysis, oxidative phosphorylation, and fatty acid oxidation. This cross-sectional study investigates the effect of different diets on exercise-induced regulation of sirtuins. SIRT1 and SIRT3–SIRT5 were measured in blood from omnivorous, lacto-ovo vegetarian, and vegan recreational runners (21–25 subjects, respectively) before and after exercise at the transcript, protein, and enzymatic levels. SIRT1, SIRT3, and SIRT5 enzyme activities increased during exercise in omnivores and lacto-ovo vegetarians, commensurate with increased energy demand. However, activities decreased in vegans. Malondialdehyde as a surrogate marker of oxidative stress inversely correlated with sirtuin activities and was elevated in vegans after exercise compared to both other groups. A significant negative correlation of all sirtuins with the intake of the antioxidative substances, ascorbate and tocopherol, was found. In vegan participants, increased oxidative stress despite higher amounts of the antioxidative substances in the diet was observed after exercise.

## 1. Introduction

Sirtuins are NAD+ (Nicotinamide adenine dinucleotide)-dependent deacylases that regulate mitochondrial energy metabolism as well as a cellular response to stress [1,2]. In mammals, seven different sirtuins (SIRT1–SIRT7) with specific subcellular localization and enzymatic reactions are known [3]. SIRT1, SIRT6, and SIRT7 are nuclear enzymes; SIRT3, SIRT4, and SIRT5 are located in mitochondria; while SIRT2 is the only cytosolic sirtuin. Additionally, a subcellular shift of SIRT1 and SIRT3 into the cytosol was described under specific conditions [4]. The enzymatic reactions catalyzed by sirtuins are either a NAD+-dependent deacylation of lysine residues or a NAD+-dependent ADP (Adenosine diphosphate) -ribosylation of lysine residues [5]. The most common enzymatic reaction of sirtuins is deacylation while SIRT5 is predominantly a desuccinylase, demalonylase, and deglutarylase [6]. The acetylation level of the mitochondrial proteome is 65% [7] which makes these proteins amenable to regulation by sirtuins. Blood levels of sirtuins were shown to correlate with several organ dysfunctions like coronary heart disease in obese patients [8,9,10] or type 1 and type 2 diabetes [11].

The nutritional state may influence sirtuin activity. Caloric restriction has been linked to longevity and protection from age-related diseases via sirtuins [12,13]. A promoting effect of polyphenols on sirtuin activity [14,15] and an activating effect of resveratrol have been described [15]. As polyphenols are phytochemicals, we hypothesized that a plant-based diet positively affects sirtuin activity. Plant-based diets such as vegetarian (predominant consumption of plant-based foods) and vegan (exclusive consumption of plant-based foods) nutrition are high in a variety of polyphenols. A lacto-ovo vegetarian diet based on a broad variety of foods is supposed to protect from obesity, type 2 diabetes, hypertension, cardiometabolic disorders, and cancer [16,17,18,19,20,21]. On the other hand, plant-based diets, if not supplemented, are low in vitamin B_12_, which could also have modulating effects on sirtuin activity [22]. Ghemrawi and colleagues described a decreased SIRT1 expression due to a lack of vitamin B_12_ [22]. These observations prompted us to compare sirtuins in blood from vegans, lacto-ovo vegetarians, and omnivores undergoing short-term exercise.

During physical exercise, energy demand increases, especially in skeletal and cardiac muscle; the energy demand of the heart, for example, may vary 10-fold. To maintain metabolic homeostasis, energy production must increase commensurate with increased energy demand during exercise. Mitochondrial energy production is regulated passively via substrate (ADP) saturation and actively via activation of ATP synthase (complex 5 of the respiratory chain) as previously described [23]. Furthermore, the Krebs cycle enzymes are under the control of calcium [24] which increases in response to exercise. Furthermore, intense short-term exercise induces oxidative stress. Sirtuins are regulators of both energy production and antioxidative response [1,25]. SIRT3 activates complex 1 [26,27], complex 2 [28], complex 3 [27], complex 4 [29], and complex 5 [30] of the respiratory chain. Mitochondrial biogenesis during exercise is regulated by SIRT3 [31] as well as SIRT1 [32]. In addition, SIRT1 stimulates mitochondrial biogenesis and oxidative phosphorylation via PGC-1α (Peroxisome proliferator-activated receptor gamma coactivator 1-alpha) [33], while fatty acid oxidation is inhibited by SIRT4 [34]. Fusion and fission of mitochondria are adaptive processes during exercise [35]. Additionally, sirtuins seem to regulate antioxidative responses, for example, by directly activating SOD2 (Superoxide dismutase 2) or inducing the glutathione system (SIRT3) [36,37,38]. Hence, a better understanding of the regulation of human energy metabolism and antioxidative response by sirtuins may offer a new approach to exercise physiology.

In recent years, exercise has been shown to activate sirtuins, namely SIRT1 and SIRT3 [32]. Furthermore, an investigation by Villanova and colleagues showed that sirtuin activity might be upregulated by physical exercise [39]. Suwa et al., as well as Covington et al., also described the upregulation of sirtuins after exercise [40,41]. A similar result was observed in rats with exercise training on treadmills resulting in activation of SIRT1 signaling pathways [42]. Zhuang et al. additionally showed transactivation of SIRT1 by FOXO3 (Forkhead box O3) and p53 in response to exercise [43].

In recent years, plant-based diets like lacto-ovo vegetarian and vegan nutrition have become increasingly popular. As diet may affect sirtuins, both basal sirtuin function, as well as sirtuin response to exercise, may be affected by diet. We hypothesize that diet may not only influence basal sirtuin levels, but the sirtuin response to exercise performance as well. While some sirtuin studies were performed in skeletal or cardiac muscle tissue, our study uses peripheral blood. This non-invasive technique allows longitudinal studies. Blood is directly affected by nutrition, so it may be more appropriate to use blood than solid tissues when studying the impact of diet on sirtuins in omnivores, lacto-ovo vegetarians, and vegans before and after intense short-term physical exercise. Alterations in blood obviously do not necessarily reflect changes in heart or skeletal muscle. We assume that sirtuin activities in blood mainly reflect activities in peripheral blood mononuclear cells (PBMCs).

## 2. Materials and Methods

### 2.1. Participants

This study is part of a cross-sectional study described previously [44,45,46]. Participants (age 18–35 years) were recruited from the general population in Hannover, Germany, by advertisements. Participants were pre-selected via screening questionnaires according to the following inclusion criteria: omnivorous, lacto-ovo vegetarian, or vegan diet for at least half a year; body mass index (BMI) between 18.5 and 25.0 kg/m^2^; and regular running exercise 2 to 5 times per week. The following criteria led to exclusion: any cardiovascular, metabolic, or malignant disease; diseases regarding the gastrointestinal tract; pregnancy; nutrient intolerances; and addiction to drugs or alcohol. Participants were matched according to age and gender.

Ethical approval was granted by the Ethics Committee at the Medical Chamber of Lower Saxony (Hannover, Germany; 12/2017). The study was conducted in accordance with the Declaration of Helsinki. All participants gave their written informed consent before recruitment. This study is registered in the German Clinical Trial Registry (DRKS00012377).

### 2.2. Methods

All subjects underwent a sports-medical examination [46]. In addition, a 24 h dietary recall was conducted (food and beverages consumed in the last 24 h, including antioxidants, polyphenols, caffeine, vitamin B_12_). Subsequently, an incremental exercise test was performed on a bicycle ergometer (Excalibur, Lode B.V., Groningen, Netherlands) until voluntary exhaustion. Participants were asked not to perform any strenuous activities one day before and on the same day of the exercise test, and to maintain their usual diet. The test started with a warm-up phase (6 min at 50 W) and increased by 16.7 W every minute. For maximum performance, the body weight-related power output (W/kg BW) and time to exhaustion (s) were determined. The subjects were verbally motivated to ensure that the subjects achieved their maximum performance [46]. Before and after the exercise test (pre and post), venous blood samples were collected, aliquoted immediately, and stored at −80 °C.

#### 2.2.1. Sample Preparation

Then, 2 mL EDTA-blood from each survey participant was taken and 500 µL of the blood sample was transferred into RNAprotect Animal Blood Tubes (Qiagen, Hilden, Germany). These samples were used for RNA isolation. The remaining 1.5 mL of blood was centrifuged at 3300× *g* for 3 min. We collected the blood plasma and used it for analysis of sirtuins.

For malondialdehyde (MDA) analysis, plasma samples (500 µL) were aliquoted in 1.5 mL Eppendorf Tubes^®^ (Eppendorf AG, Hamburg, Germany) immediately after centrifugation (4 °C, 1620× *g*) and measured by validated stable-isotope dilution gas chromatographic-mass spectrometric (GC-MS) methods [47,48] (for details see [49]).

#### 2.2.2. RNA Isolation and qRT-PCR

RNA was isolated with RNeasy Protect Animal Blood Kit (Qiagen) according to the product protocol. RNA quantity and quality were assessed using a NanoDrop™ 2000 measuring E_260_ and ratio E_260_/E_280_. The isolated RNA was reverse transcribed to cDNA with the Omniscript RT Kit (Qiagen). Real-time PCR of different cDNA samples was carried out with SYBR green on a 7900 HT fast real-time PCR system (Applied Biosystems, Darmstadt, Germany). Used primers are shown in Appendix A. Relative changes in the mRNA expression were calculated according to Vandesompele et al. [50] with SUPT20H as a reference for relative quantification.

#### 2.2.3. Sirtuin Activity Assay

SIRT1 and SIRT3 deacetylase activities and SIRT5 desuccinylase activity were determined by using SIRT1, SIRT3, and SIRT5 fluorometric drug discovery assay kits (Enzo Life Science, Lausen, Switzerland). To ensure that enzyme capacity (maximal in vitro enzyme activity under substrate saturation) was measured we added a surplus of NAD+ to our assays. We followed the manufacturer’s protocol with plasma samples diluted 1:5 in HEPES buffer (110 mM NaCl, 2.6 mM KCl, 1.2 mM KH_2_PO₄, 1.2 mM MgSO_4_ × 7H_2_O, 1.0 mM CaCl_2_, 25 mM HEPES) and lysed by sonification for 10 s at 20 kHz with an amplitude of 75%. For normalization of the measured SIRT-activity signals, we measured total protein concentration of the analyzed samples lysed in HEPES buffer with the Pierce™ BCA Protein Assay Kit (Thermo Fisher Scientific, Waltham, MA, USA).

We are aware that these tests were originally developed for screening modulators of sirtuin activities and substrate-specificity is limited.

### 2.3. Data Analysis and Statistical Methods

Statistical analyses were performed using SPSS software (IBM SPSS Statistics 24.0; Chicago, IL, USA) and GraphPad Prism 7.02 (GraphPad Software Inc., San Diego, CA, USA). Results are shown as median (+max/−min). First, normal distribution was checked by using the Kolmogorov–Smirnov test. If data were normally distributed, one-way ANOVA and two-way ANOVA were used to evaluate differences between the three diet groups. For data with non-parametric distribution, a Kruskal–Wallis test was performed. Additionally, if there were significant differences between the groups, a post hoc test with Bonferroni correction was conducted (Dunn´s multiple comparison test). Moreover, the Chi-squared test was used to compare the differences between the frequency distribution of the three groups. Associations between parametric data were computed via Pearson and non-parametric data were computed via Spearman´s rho correlation. Lastly, *p*-values ≤ 0.05 were regarded as statistically significant.

To analyze the nutrient intake of the 24 h dietary record, the nutrition organization software PRODI^®^ (Nutri-Science GmbH, Freiburg, Germany) was used.

## 3. Results

In total, 76 healthy male and female omnivorous (OMN), lacto-ovo vegetarian (LOV), and vegan (VEG) recreational runners aged between 18 and 35 years were included in the study [46]. However, in five subjects no analyses could be performed due to a failure to collect blood or (pre-) analytical errors (Appendix A). Details of the study population are summarized in Table 1.

As previously described [45,46], all subjects were adequately supplied with vitamin B_12_, D, and iron and did not differ significantly regarding maximum power output during the exercise test (OMN: 4.15 ± 0.48, LOV: 4.20 ± 0.47, VEG: 4.16 ± 0.55 Watt/kg BW) and time to exhaustion (OMN: 1199 ± 177, LOV: 1197 ± 183, VEG: 1187 ± 237 s).

### 3.1. Sirtuin Activity

We analyzed the sirtuin capacity (under substrate saturation) in vitro in the blood of 71 participants before and after exercise (Table 1). There were no significant differences in basal sirtuin capacity levels of all sirtuins between the three nutritional groups and no gender differences were observed. We detected a significant increase of SIRT1 enzyme capacity in omnivores (Figure 1A) with a median capacity of 0.0038 U/μg protein (+0.0034/−0.0016) before and 0.0045 U/μg (+0.0126/−0.0021) after exercise. The induction of the SIRT1 capacity in response to exercise in lacto-ovo vegetarians was not significant with activity levels of 0.0049 U/μg (+0.003/−0.0028) before and 0.0055 U/μg (+0.0083/−0.0026) after exercise. A similar result was observed for samples from vegan participants. Here, the values before and after exercise were unchanged with median SIRT1 capacity levels of 0.0038 U/μg (0.006/−0.0018) before and 0.0042 U/μg (+0.0026/−0.0024) after exercise (Figure 1A).

The results were similar for omnivores in the case of SIRT3 (before (0.0105 U/μg (+0.0145/−0.0032)) and after (0.0135 U/μg (+0.0209/−0.0052)) exercise, *p* < 0.05) and SIRT5 (before (0.0007 U/μg (+0.0004/−0.0006)) and after (0.0009 U/μg (+0.0006/−0.0006)) exercise, *p* < 0.05). For lacto-ovo vegetarians, changes from 0.0115 U/μg (+0.0032/−0.0073) before to 0.0139 U/μg (+0.0064/−0.0088) after exercise for SIRT3 (Figure 1B) and from 0.0004 U/μg (+0.0005/−0.0002) before to 0.0005 U/μg (+0.0008/−0.0002) after exercise for SIRT5 were not significant (Figure 1C). SIRT3, as well as SIRT5, levels of sirtuin capacity decreased in samples of vegan participants. For SIRT3 a reduction of ~10% to 0.0090 U/μg (+0.0123/−0.0023) after exercise and from 0.00058 U/μg (+0.0007/−0.00045) before to 0.00046 U/μg (+0.00098/−0.0004) after exercise was observed for SIRT5.

Since we observed an altered result in participants with a vegan diet, we reanalyzed our data for sirtuin capacity with a paired analysis approach to detect intraindividual alterations within single participants. Therefore, we subtracted the sirtuin capacity before exercise from the sirtuin capacity after exercise.

In addition, the SIRT1 capacity was reduced in response to exercise in vegan participants. While there was an induction of 0.001–0.002 U/μg protein in omnivores and lacto-ovo vegetarians, the SIRT1 capacity in vegans was reduced by ~0.0007 U/μg protein. There was a significant difference compared to omnivores and lacto-ovo vegetarians as well (Figure 2A).

For SIRT3, a similar result was observed (Figure 2B). In omnivores, we detected an induction by 0.003 U/μg protein after exercise and an increase of 0.002 U/μg protein in lacto-ovo vegetarians. For samples of vegan participants, we observed a slight decrease of 0.0005 U/μg protein. The vegan group differed again significantly from the omnivorous and lacto-ovo vegetarian group.

SIRT5 showed a slightly different result (Figure 2C). Similar to SIRT1 and SIRT3, omnivores showed an increase in enzyme activity by 0.00016 U/μg protein. For vegan participants, a significantly different reduction by 0.0004 U/μg protein was observed. In contrast to the results of SIRT1 and SIRT3, we detected only a small induction by 0.00002 U/μg protein in the lacto-ovo vegetarian group, resulting in no significant difference between vegan and lacto-ovo vegetarian participants in SIRT5.

Although the change in sirtuin capacity was likely caused by altered posttranslational modifications, we examined possible changes at gene expression levels of the analyzed sirtuins.

We measured the relative expression levels of SIRT1, SIRT3, SIRT4, and SIRT5. Basal levels before exercise were not different between groups and there were no gender differences. The changes in expression levels were calculated similarly to the changes in sirtuin capacity (under substrate saturation) before and after exercise. No significant change in gene expression was detected for any of the analyzed sirtuins SIRT1, SIRT3, SIRT4, and SIRT5 (Figure 3A–D). The overall distribution of relative expression levels of vegan participants was similar to the omnivore and vegetarian groups.

A representative set of samples of each group was tested and in the examined samples no differences between the groups were detectable.

### 3.2. Oxidative Stress

Since measuring the activity of active reactive oxygen species (ROS) detoxification enzymes in frozen blood samples is difficult due to the instability of the enzymes, we examined the levels of malondialdehyde (MDA) as a surrogate marker for oxidative stress [49]. We detected a general increase of MDA after exercise in all three study groups (OMN: +9%; LOV: +24%; VEG: +15%). Interestingly the levels of MDA after exercise were significantly higher in individuals with significantly lower sirtuin enzyme activity (VEG) (Figure 4B). Furthermore, the increase of MDA levels in each participant was significantly higher in LOV and VEG groups after exercise compared to OMN (Figure 4C). Since there was no difference in basal levels, it seems to be an exercise-induced effect.

In contrast to MDA levels, the nitrate levels as a marker for nitric oxide (NO) levels were not affected by exercise [43]. Although exercise is generally considered to increase NO formation in blood vessels by elevating endothelial NO synthase, the study population showed no statistically significant changes in plasma nitrate and nitrite concentrations [49].

### 3.3. Correlations

In an attempt to find possible explanations for the differences between vegan participants and the omnivorous and lacto-ovo vegetarian groups we correlated the capacities of SIRT1, SIRT3, and SIRT5 with different parameters from the 24 h dietary recall. We tested potential correlations between sirtuin capacities and several substances potentially influencing sirtuin activities. We did not detect correlations of sirtuin capacity with caffeine intake, blood insulin levels, blood glucose levels, and active and total vitamin B_12_ concentrations in blood. Additionally, we tested possible correlations of polyphenolic and flavonoid substances from the dietary recall, also without significant correlations. Furthermore, we found no correlations between sirtuin capacities and the exercise intensity of probands and since sirtuins are responding to caloric restriction, correlations of sirtuin capacities with the caloric intake. All of these correlations were not significant (*p*-values > 0.05) (Appendix A).

We found significant inverse correlations of sirtuin activities with the intake of the antioxidative substances tocopherol and ascorbate. Tocopherol showed a significant correlation (*p* < 0.05, *r* = 0.27) with all three analyzed sirtuin enzyme capacities (Figure 5A–C). For ascorbate, the correlations were similar but only in the case of SIRT1 was it statistically significant (*p* = 0.042, *r* = 0.28) (Figure 5D), while the correlations showed low but not significant *p*-values for SIRT3 (*p* = 0.061, *r* = 0.25) (Figure 5E) and SIRT5 (*p* = 0.148, *r* = 0.17) (Figure 5F).

We calculated ascorbate and tocopherol intake within the 24 h recall period in the study groups OMN, LOV, and VEG. In vegan participants, we found increased intake of ascorbate as well as tocopherol (Figure 6), resulting in previously described correlations.

Additionally, we tested a putative correlation of sirtuin activity levels with MDA levels as a marker of oxidative stress. We demonstrated a significant correlation of SIRT1 enzyme capacity with the MDA levels within the whole study population (*p* = 0.032, *r* = 0.213) (Figure 5G). Surprisingly, the correlations of SIRT3 (*p* = 0.208) and SIRT5 (*p* = 0.163) enzyme capacities with MDA levels were not significant (Figure 5H–I). This is probably due to low linear regression levels (SIRT3: *r* = 0.137; SIRT5 *r* = 0.159)

Several subjects took dietary supplements (OMN: 38.5, LOV: 34.6, VEG: 62.5%) as potential confounders. Commonly consumed supplements were magnesium (OMN: 23.1, LOV: 15.4, VEG: 16.7%), iron (OMN: 7.69, LOV: 11.5, VEG: 16.7%), vitamin B_12_ (OMN: 19.2, LOV: 15.4, VEG: 50%), and vitamin D (OMN: 23.1, LOV: 3.85, VEG: 20.8%).

## 4. Discussion

To the best of our knowledge, this is the first investigation of the impact of dietary habits on the exercise-induced regulation of sirtuins in human peripheral blood.

Sirtuins are known to be linked to nutrition. The first evidence of sirtuins as functional markers in blood was published by Tarantino et al. [8] with SIRT4 showing an inverse correlation to obesity. Alterations of sirtuins were described under caloric restriction with induction of sirtuin expression and activity, as reviewed by Kapahi et al. [51]. Furthermore, it was reported that sirtuin activities can be altered by glucose supply [52]. We are aware that blood sirtuins may not necessarily reflect sirtuin function in tissue, however, we think that blood levels are a reasonable surrogate parameter for tissue levels.

We hypothesized that sirtuins are altered in omnivores, lacto-ovo vegetarians, and vegans as previously observed in different animals [42,43]. In our study, we compared omnivores to lacto-ovo vegetarians and vegans before and after intense short-term exercise. For sirtuins SIRT 1, SIRT3, SIRT4, and SIRT5 in blood, no differences could be observed at the gene expression level before exercise (results not shown). At the basal enzyme level, there were no differences for sirtuins SIRT1, SIRT3, and SIRT5. For SIRT4, no commercial enzyme assay was available; therefore, the measurement was done only at the gene expression level. In a pilot study, basal sirtuin activities at recruitment were compared to enzyme activities just before exercise (*n* = 5), no differences were found.

Increased energy demand during exercise has to be met by increased flux at the levels of glycolysis, the Krebs cycle, fatty acid oxidation, and the mitochondrial respiratory chain (oxidative phosphorylation). Increased capacities of sirtuins SIRT1 and SIRT3 as observed in our study in omnivores, and to a lesser (non-significant) extent in lacto-ovo vegetarians, during physical exercise resulted in activation of these pathways [27,29,30,31,33,34,53,54,55]. Previous studies in different animals showed induction of SIRT1 levels after exercise as well [40,43]. In our study, the capacity of SIRT4, an important regulator of fatty acid oxidation [34], could not be measured in the absence of an adequate assay. In vegans, sirtuin capacities decreased or remained unchanged during exercise. The reasons for and consequences of these results are not clear. Downregulation of sirtuin capacities may be due to exercise-induced oxidative stress, despite high dietary intake of antioxidants in vegans. On the other hand, reduced sirtuin capacity may result in oxidative stress with the consumption of antioxidants. It could be speculated that a decrease of sirtuin capacities may result in energy deficiency in skeletal and heart muscle during exercise and there are indications of such correlations [56]. However, it is not clear if alterations of sirtuins in blood translate into sirtuin modification in muscle. In our study collective, maximal power output and time to exhaustion were not affected by dietary habits. Furthermore, only in vitro enzyme capacities under substrate saturation were measured, which do not necessarily reflect actual in vivo activities. Whether this leads to clinical symptoms or subclinical energy deficiency in tissues is still unknown.

Basal levels of enzyme capacities did not differ between the groups (Figure 1) which may suggest a similar nutrition level for all study participants. SIRT5 capacity was somewhat lower in lacto-ovo vegetarians compared to the other groups, though not significantly. This may indicate reduced protein intake in vegetarians since SIRT5 is an important regulator of the urea cycle, where toxic ammonia from protein degradation is converted to urea. Indeed, the protein intake in the 24 h dietary recall was slightly reduced in vegetarians in a non-significant manner (*p* = 0.22).

We previously reported a correlation between ROS levels after treatment with antioxidants and sirtuin activities [25]. This prompted us to correlate the dietary intake of the antioxidants tocopherol and ascorbate assessed by the dietary recall with sirtuin levels. We found a negative correlation between these compounds with sirtuin capacities in all participants.

Increasing evidence suggests that sirtuins play an important role in stress responses [1]. In particular, SIRT3 is involved in the cellular response to oxidative stress by deacetylating and activating the SOD2 [36,57]. Furthermore, SIRT3 affects ROS detoxification by inducing the glutathione system as well as the thioredoxin system [38]. This may protect organs from exercise-induced mitochondrial ROS production [58].

Reduced SIRT1 and SIRT3 enzyme activities have a variety of different cellular consequences. Energy metabolism, especially the mitochondrial respiratory chain, is downregulated [59] as well as the antioxidative response. SOD2 is a target of SIRT1 as well as SIRT3 and is the main ROS detoxification enzyme in mitochondria [36,37]. Only a few publications give evidence for a direct effect of antioxidative substances in humans. Some studies suggest that altered ROS levels act as effectors on sirtuin activity in response to high levels of antioxidants during exercise [60].

In this study, exercise led to increases in plasma MDA concentrations in the LOV (+24%) and VEG (+15%) groups, suggesting higher exercise-induced ROS levels [49]. This may be an effect of reduced SIRT1 and SIRT3 enzyme activity, hence a lack of sirtuin-dependent activation of SOD2 and catalase. However, sirtuin response to exercise was different in LOV and VEG subjects which argues against a causal relationship between sirtuins and MDA. Since measuring enzyme activity of antioxidative enzymes is challenging in frozen blood samples, we think that the observed increase of MDA as a marker for general oxidative stress may indicate the importance of sirtuins in ROS defense. Given the growing evidence for the effect of beneficial ROS levels on lifespan extension [61] the observed levels of ROS and their subcellular localization are an interesting field for future investigation in people practicing LOV and VEG diets.

In contrast to ROS levels, physical exercise did not cause statistically significant changes in plasma nitrate and nitrite concentrations, suggesting that the chosen exercise protocol (incremental stress test on a bicycle ergometer) did not alter the metabolism (e.g., reduction of nitrate to nitrite), although different observations can be found in the literature [61].

Our study has some limitations. We measured sirtuins in blood not necessarily reflecting sirtuin function in solid tissues. Sirtuins measured in blood presumably reflect the activity and expression in peripheral blood mononuclear cells [61. To our knowledge, there is no evidence that sirtuin activity in blood represents sirtuins leaking out of solid organs (skeletal and cardiac muscle). It has previously been shown that sirtuin levels in blood correlate with different organ dysfunctions like coronary heart disease in obese patients [8,9,10,62] or type 1 and type 2 diabetes [11]. Exercise leads to metabolic stress in different organs and it would not ethically be sound to take biopsies from multiple organs in humans. Therefore, blood levels of sirtuins were used as surrogate parameters of sirtuin function in tissues. Another limitation is the specificity of the tests applied to measure sirtuin activities. They were originally developed to assay modulators of sirtuins, and their specificity is limited. A way around this issue would be to use specific antibodies or inhibitors.

Sirtuin measurements in blood samples are of special interest for future clinical studies in probands or patients undergoing pharmacological or dietary treatment. For example, in multiple sclerosis sirtuins may upregulate mitochondrial melatonin synthesis [63]. Patients with inborn errors of metabolism are of special interest for future studies, as many of them are treated with a special diet.

## 5. Conclusions

In conclusion, we showed that sirtuins can be measured in human blood at enzyme and gene expression levels. Basal enzyme capacities and gene expression levels of sirtuins SIRT1, SIRT3, and SIRT5 were not influenced by dietary habits (omnivores, lacto-ovo vegetarians, and vegans); at the gene expression level, there was no impact of diet on SIRT4. While enzyme capacities of SIRT1 and SIRT3 were upregulated during exercise in omnivores, and to a lesser extent in lacto-ovo vegetarians, as a reflection of increased energy demand, enzyme capacities of sirtuins SIRT1, SIRT3, and SIRT5 were downregulated in blood from vegans. This may be related to increased oxidative stress as shown by elevated MDA levels despite higher intake of the antioxidants tocopherol and ascorbate as judged by dietary recalls. Whether these changes are of clinical relevance, remains to be elucidated.

Measurement of sirtuins in the blood is a non-invasive tool to study metabolic regulation in probands and patients undergoing pharmacological and/or dietary manipulation.

## Figures and Tables

**Figure 1 nutrients-12-01004-f001:**
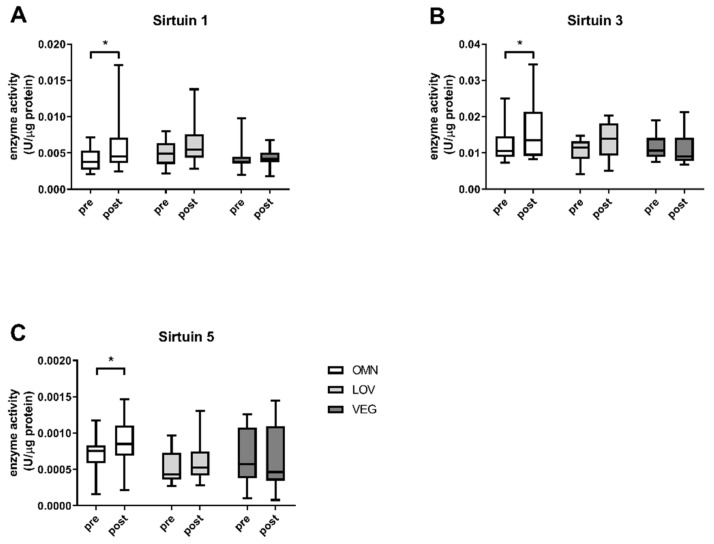
Absolute enzyme capacities (under substrate saturation) of sirtuins SIRT1, SIRT3, and SIRT5. The figure shows the enzyme activities before (pre) and after (post) exercise in the three analyzed study groups of omnivores (OMN), lacto-ovo vegetarians (LOV), and vegans (VEG) for the sirtuins SIRT1 (**A**), SIRT3 (**B**), and SIRT5 (**C**). Data are shown as median ± quartiles and extrema; *n* = 21–25; statistical analysis with Kruskal–Wallis test and Dunn´s multiple comparison test; * *p* < 0.05.

**Figure 2 nutrients-12-01004-f002:**
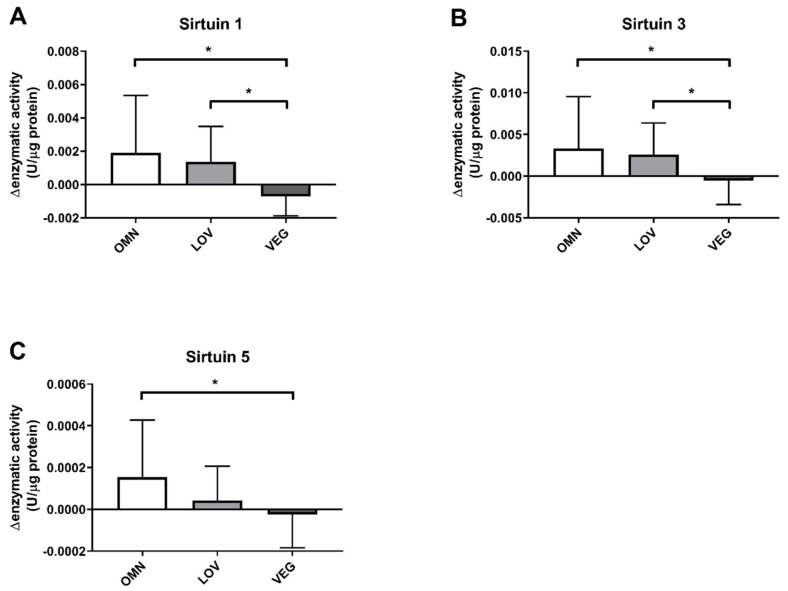
Changes of enzyme capacity (under substrate saturation) of sirtuins SIRT1, SIRT3, and SIRT5 after exercise. The response of sirtuins to exercise was calculated as the difference of enzyme capacities before (pre) and after (post) exercise. Sirtuins in the three study groups of omnivores (OMN), lacto-ovo vegetarians (LOV), and vegans (VEG) are shown: SIRT1 (**A**), SIRT3 (**B**), and SIRT5 (**C**). Data are shown as mean difference ± SD; *n* = 21–25; statistical analysis with Kruskal–Wallis test and Dunn´s multiple comparison test; * *p* < 0.05.

**Figure 3 nutrients-12-01004-f003:**
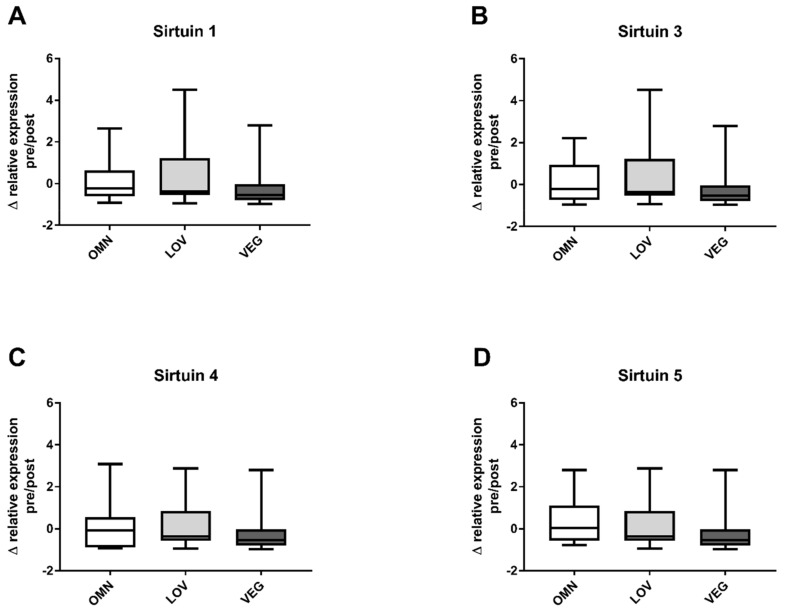
Changes in the relative expression of sirtuins SIRT1, SIRT3, SIRT4, and SIRT5 after exercise. The figure shows the differences of relative expression before (pre) and after (post) exercise in the three analyzed study groups of omnivores (OMN), lacto-ovo vegetarians (LOV), and vegans (VEG) for the analyzed sirtuins SIRT1 (**A**), SIRT3 (**B**), SIRT4 (**C**), and SIRT5 (**D**). Data are shown as median ± quartiles and extrema; *n* = 21–25; statistical analysis with Kruskal–Wallis test.

**Figure 4 nutrients-12-01004-f004:**
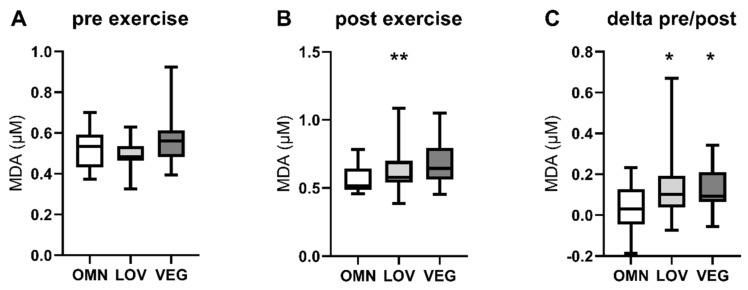
Levels of malondialdehyde (MDA) as a marker for oxidative stress pre, post, and pre/post exercise. The figure shows the levels of blood MDA before (pre) (**A**) and after (post) (**B**) exercise in the three analyzed study groups omnivores (OMN), lacto-ovo vegetarians (LOV), and vegans (VEG) (**B**). The change of MDA levels in blood calculated as pre/post difference is also shown (**C**). Data are shown as median ± quartiles and extrema; *n* = 21–25; statistical analysis with Kruskal–Wallis test; * *p* < 0.05, ** *p* < 0.01.

**Figure 5 nutrients-12-01004-f005:**
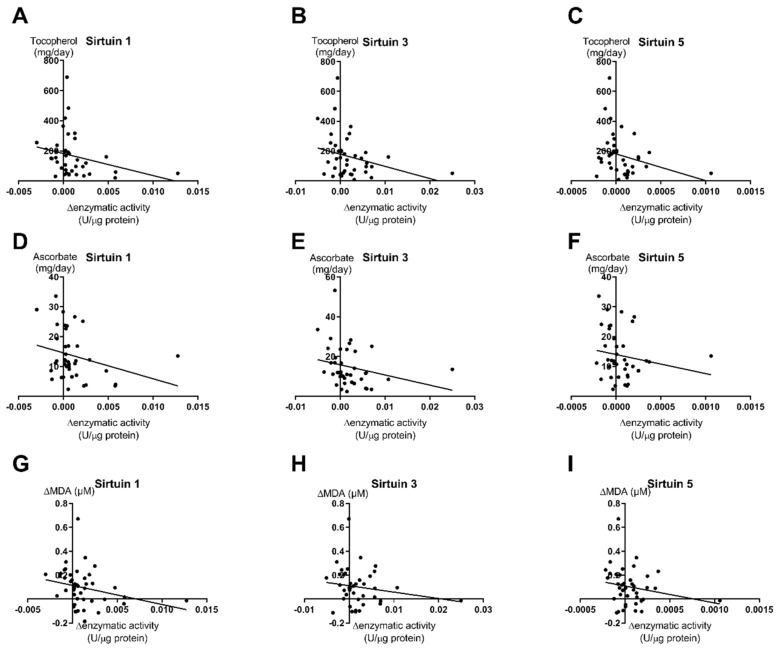
Correlations of calculated tocopherol and ascorbate intake with SIRT1, SIRT3, and SIRT5 capacity levels. The figure shows the correlations of changes in enzyme capacities (post-pre exercise) with either tocopherol (**A**–**C**), ascorbate (**D**–**F**) or MDA (**G**–**I**) for all analyzed sirtuins. For correlation analyses, all study groups were pooled (*n* = 71). Correlations for tocopherol and ascorbate with SIRT1 were statistically significant; statistical analysis with Spearman’s correlation test.

**Figure 6 nutrients-12-01004-f006:**
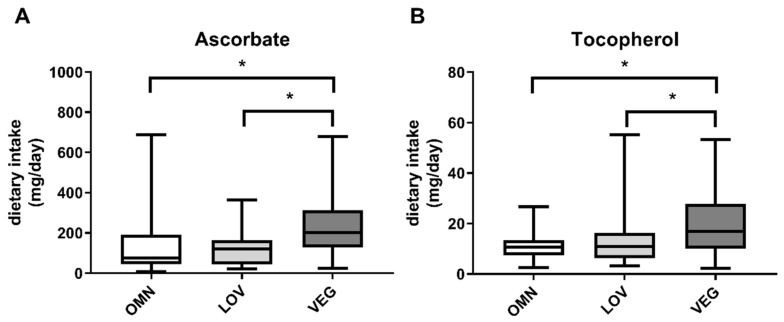
Intake of ascorbate (**A**) and tocopherol (**B**) in the three analyzed study groups of omnivores (OMN), lacto-ovo vegetarians (LOV), and vegans (VEG) during 24 h dietary recall. Data are shown as median ± quartiles and extrema; *n* = 21–25; statistical analysis with Kruskal–Wallis test and Dunn´s multiple comparison test; * *p* < 0.05.

**Table 1 nutrients-12-01004-t001:** Participant characteristics by dietary patterns of the study population.

	OMN (*n* = 25)	LOV (*n* = 25)	VEG (*n* = 21)	*p*-Value
Age (y)	27.2 ± 4.1	27.6 ± 4.4	27.2 ± 4.4	0.888 ^a^
Sex	m = 10, f = 15	m = 10, f = 15	m = 9, f = 12	0.975 ^b^
BMI (kg/m^2^)	22.3 ± 1.74	21.6 ± 1.98	22.1 ± 2.09	0.426 ^a^
LBM (kg)	54.1 ± 9.2	52.7 ± 8.9	54.6 ± 11.3	0.869 ^a^
Body fat (%)	21.4 ± 6.0	21.2 ± 5.6	20.2 ± 5.3	0.752 ^c^
Duration of diet				0.001 ^b^
< 0.5 years, *n* (%)	0 (0)	0 (0)	0 (0)	
0.5–1 year, *n* (%)	0 (0)	4 (16)	5 (24)	
1–2 years, *n* (%)	1 (4)	3 (12)	3 (14)	
2–3 years, *n* (%)	0 (0)	2 (8)	7 (33)	
>3 years, *n* (%)	24 (96)	16 (64)	6 (29)	
Smoker (%)	0	0	0	–
Training frequency per week	3.0 ± 0.9	3.2 ± 0.9	2.9 ± 0.8	0.469 ^a^
Running time per week (h)	2.7 ± 1.1	3.3 ± 1.3	2.6 ± 1.5	0.237 ^a^

OMN = omnivores, LOV = lacto-ovo vegetarians, VEG = vegans, SU = supplement users, n.s. = not significant, BMI = body mass index, LBM = lean body mass. Values are given as means ± SD or *n* (%). ^a^ Kruskal–Wallis test, ^b^
*Chi*-squared test, ^c^ one-way ANOVA.

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
