# Peer review of "Impact of Nutrition on Short-Term Exercise-Induced Sirtuin Regulation: Vegans Differ from Omnivores and Lacto-Ovo Vegetarians"

_nutrients, 2020, doi:10.3390/nu12041004_

Round 1

Reviewer 1 Report

This is a clearly written and interesting manuscript with some novel findings that may have wider implications as to the impacts of dietary intake on exercise-driven changes in mitochondrial metabolism.

-Minor addition: Sirtuins can inhibit pyruvate dehydrogenase kinase, and thereby disinhibit the pyruvate dehydrogenase comples, leading to an increased conversion of pyruvate to acetyl-CoA. As acetyl-CoA contributes to both the Kreb's/TCA cycle and oxidative phosphorylation, it would be interesting to state that this could be an important metabolic alterations that may arise following exercise in people under different dietary regimes. Recent work indicates that acetyl-CoA, being a necessary substrate for AANAT and the mitochondrial melatonergic pathway, may act to upregulate melatonin production in mitochondria, with consequences for the regulation of SOD2 and different sirtuins [e.g. Anderson et al., 2019]. This may be good to include as to possible future research directions near the end of the discussion, as a number of factors can act to regulate mitochondrial melatonin levels.

Multiple Sclerosis: Melatonin, Orexin, and Ceramide Interact with Platelet Activation Coagulation Factors and Gut-Microbiome-Derived Butyrate in the Circadian Dysregulation of Mitochondria in Glia and Immune Cells.

Anderson G, Rodriguez M, Reiter RJ.

Int J Mol Sci. 2019 Nov 5;20(21). pii: E5500. doi: 10.3390/ijms20215500.

Author Response

Point-by-point response to the comments made by the reviewers

We thank the reviewers for their kind comments and the opportunity to revise our paper. The comments and suggestions have helped to improve the manuscript substantially. In the following, each comment is addressed separately.

Reviewers’ comments:

Reviewer # 1:

This is a clearly written and interesting manuscript with some novel findings that may have wider implications as to the impacts of dietary intake on exercise-driven changes in mitochondrial metabolism.

-Minor addition: Sirtuins can inhibit pyruvate dehydrogenase kinase, and thereby disinhibit the pyruvate dehydrogenase comples, leading to an increased conversion of pyruvate to acetyl-CoA. As acetyl-CoA contributes to both the Kreb's/TCA cycle and oxidative phosphorylation, it would be interesting to state that this could be an important metabolic alterations that may arise following exercise in people under different dietary regimes. Recent work indicates that acetyl-CoA, being a necessary substrate for AANAT and the mitochondrial melatonergic pathway, may act to upregulate melatonin production in mitochondria, with consequences for the regulation of SOD2 and different sirtuins [e.g. Anderson et al., 2019]. This may be good to include as to possible future research directions near the end of the discussion, as a number of factors can act to regulate mitochondrial melatonin levels.

Multiple Sclerosis: Melatonin, Orexin, and Ceramide Interact with Platelet Activation Coagulation Factors and Gut-Microbiome-Derived Butyrate in the Circadian Dysregulation of Mitochondria in Glia and Immune Cells.

Anderson G, Rodriguez M, Reiter RJ.

Int J Mol Sci. 2019 Nov 5;20(21). pii: E5500. doi: 10.3390/ijms20215500.

Response:

Thank you for this suggestion. This issue and the corresponding reference were included in the manuscript’s discussion.

Reviewer 2 Report

Basically, the cross-sectional study presented by Potthast and co-workers attempts to reveal how diet may affect a maximal exercise-induced response of recreational runners in terms of sirtuin-mediated adaptation to oxidative stress (OS). Despite the potential interest of this work for many readers, and the relative novelty of the data presented, many aspects were omitted and many serious flaws of the study should be fixed.

Major points

One of the major problem of this paper is the fact that the enzymatic activities of SIRTs were assessed using commercially available kits. The Authors did not report any useful information about the kits used, however it is well known that many of the common commercial kits that claim to measure exclusively either SIRT1 or SIRT3 activity use variants of the same protein (please, see https://www.enzolifesciences.com/BML-AK557/fluor-de-lys-sirt3-fluorometric-drug-discovery-assay-kit/ and https://www.enzolifesciences.com/BML-AK555/fluor-de-lys-sirt1-fluorometric-drug-discovery-assay-kit/). In fact, in the reviewer's personal experience in assessing SIRT1 activity, the substrate provided with an Abcam kit is deacetylated not only by SIRT1, but also by recombinant SIRT3 (and the "native" SIRT3 was not tested yet with the kit) (exchange of emails between me and Helga Palma from Scientific Support Specialist in Abcam plc., Jun 1st, 2018). How did I solve the problem? My samples underwent immunoprecipitation before processing with the kit. An alternative approach would require a specific inhibitor of either SIRT1 or SIRT3. 

At page 3 (Materials and Methods), the Authors should expand their descriptions of the procedures followed. For example, how did the authors assess RNA quantity and quality (with the latter being a critical factor for downstream PCR-based reactions)? Furthermore, how was "MDA" assessed? Is a 2.2.4 missing? Please, be advised that MDA is often used as a standards in TBARS-measuring kits.

At page 3 (Materials and Methods), the Authors state that "EDTA-blood" was used, even though some lines below the 2.2.3 paragraph reports that "serum samples" were processed for subsequent analyses. Please, correct this or elaborate better. Moreover, SIRTs are known to be endocellular enzymes with different and specific localizations. Why do the Authors believe that serum levels of SIRTs may somehow reflect what is found in blood cells or in other tissues (e.g., muscles)?

Statistics should also be improved. The Authors state that a One-Way ANOVA-based approach was used, however it is clear that the study design required repeated measures on the same individuals (before and after). Therefore, I strongly suggest to implement the statistics with a factorial ANOVA for repeated measures (pre-post as the repeated factor for analyzing intraindividual differences). That would also provide more robust responses to some gaps in the results and discussion (e.g., page 7, lines 223-224).

At page 4 (Results), a very intriguing result is given (no gender-difference could be found", without investigating further. To me, it requires a little more information or references. Beyond the [41] reference, is there any other literature support that reported convincing evidence of no gender-dependent differences observed in a maximum power output during incremental exercise test on a bycycle ergometer until exhaustion? This looks important, as it might reveal a possible problem with the maximal exercise test performed.

Another thing that captured the reviewer's attention was the table 1. A careful look at the the two columns LOV and VEG revealed that the two groups had not been similarly engaged with the diet regimens. In fact, many more participants had a very long story (>3 yr) with LOV diet, with respect to what seen in VEG group. This could reveal a selection bias. Maybe, the Authors should try to re-analyze data stratifying results for duration of diet. This could provide interesting clues that would be useful for expand discussion.

In the Results section (page 4), some surprising "significant" differences are reported (e.g., lines 163-165). With similar means and SDs, it would appear that those differences might be due due to chance alone (i.e., null hypothesis true). However, the Authors chose not to show such data in figures as means±SDs. Instead, data were shown as meadian±quartiles. This suggests that these data were not analyzed through parametric approaches. Therefore, in the text means±SDs should not be reported, as it may be misleading.

Data presented in the figures included basal levels only for SIRT activities. This could obscure other observations that may be of interest. For example, it would be interesting to see the basal levels of SIRT mRNAs, because this could reveal intriguing differences among groups.

With data stratification based on duration of diet regimen, it may be interesting to correlate changes after the acute exercise bout. In fact, it may be observed a differential response of relatively new-LOV (or relatively new-VEG) to the ergometric performance, with respect to the subjects with longer history of those diets. This may provide novel informations to the Authors and the readers.

Some considerations in the discussion seem to be not adequately supported by data. In particular, at page 10, the Authors state that "in vegans, sirtuin capacities decreased or remained unchanged during exercise", and this was explained reasoning about the presumable "decrease in energy deficiency in skeletal/heart during exercise". First of all, the parallel between what is found in the blood and in tissues not that obvious. Secondly, if that were true, how do the Authors explain the fact that the maximal power output and the time to exhaustion was not significantly different when LOV and VEG were compared?

More importantly, MDA was found to increase post-exercise both in LOV and VEG, even though SIRT1 and SIRT3 were observed to respond only in LOV. This may suggest that diet does not control MDA levels post-exercise via SIRT regulation. The Authors did not discuss this at all, and in the reviewer's opinion, this cannot be dismissed sic et simpliciter.

Minor points

As well known, the adaptation of energy supply to exercise is strictly dependent on mitochondrion morphological and functional re-shaping. In this context, the regulation of mitochondrial biogenesis and overall dynamics (fusion, fission, and mitophagy) is critically involved in the exercise-induced metabolic rewiring. The authors almost completely omitted this in their introduction. This should be fixed.  

At page 2 (Introduction), the Authors should expand briefly the part regarding the role of sirtuins in the cellular response to OS, because many key roles of SIRT1 and 3 are omitted (e.g., SIRT3-dependent direct activation of SOD2, SIRT1/FOXO-mediated SODs response to OS etc.)

In the Introduction, the Authors claim that "...literature regarding the influence of exercise performance on sirtuin activity in humans is scarce". A simple and quick search for "exercise sirtuins human" in NCBI returns 50 reviews and 121 items published over the past 10 years. Therefore, I strongly suggest to re-write that statement re-orienting the phrase towards the real novelty of the study, that is the diet-dependent changes.

At page 10 (Discussion), the Authors state that "no differences could be observed at the gene expression level before exercise". This was not shown.

Author Response

Point-by-point response to the comments made by the reviewers

We thank the reviewers for their kind comments and the opportunity to revise our paper. The comments and suggestions have helped to improve the manuscript substantially. In the following, each comment is addressed separately.

Reviewers’ comments:

Reviewer # 2:

Basically, the cross-sectional study presented by Potthast and co-workers attempts to reveal how diet may affect a maximal exercise-induced response of recreational runners in terms of sirtuin-mediated adaptation to oxidative stress (OS). Despite the potential interest of this work for many readers, and the relative novelty of the data presented, many aspects were omitted and many serious flaws of the study should be fixed.

Major points

One of the major problem of this paper is the fact that the enzymatic activities of SIRTs were assessed using commercially available kits. The Authors did not report any useful information about the kits used, however it is well known that many of the common commercial kits that claim to measure exclusively either SIRT1 or SIRT3 activity use variants of the same protein (please, see https://www.enzolifesciences.com/BML-AK557/fluor-de-lys-sirt3-fluorometric-drug-discovery-assay-kit/ and https://www.enzolifesciences.com/BML-AK555/fluor-de-lys-sirt1-fluorometric-drug-discovery-assay-kit/). In fact, in the reviewer's personal experience in assessing SIRT1 activity, the substrate provided with an Abcam kit is deacetylated not only by SIRT1, but also by recombinant SIRT3 (and the "native" SIRT3 was not tested yet with the kit) (exchange of emails between me and Helga Palma from Scientific Support Specialist in Abcam plc., Jun 1st, 2018). How did I solve the problem? My samples underwent immunoprecipitation before processing with the kit. An alternative approach would require a specific inhibitor of either SIRT1 or SIRT3. 

Response:

Thank you for addressing this point. We are aware of the specificity issue with the commercial test kits (we used Enzo-kits) which were designed to study modulators of sirtuin activities. Immunoprecipitation via antibodies or specific sirtuin inhibitors would be an asset. This has now been mentioned both in the methods and discussion sections as a clear limitation.

SIRT 3 is the only mitochondrial sirtuin with deacylase activity, SIRT 1 the only nuclear sirtuin with deacylase activity. In pilot experiments we isolated mitochondria from the nuclear fraction. SIRT3 activity was predominantly found in mitochondria, SIRT1-activity predominantly in the nuclear fraction. This is some indirect evidence that the assays predominantly analyze SIRT 3 and SIRT1, respectively.

Cytosol/Nuclear       Mitochondria

SIRT1            41800 RFU               7300 RFU

SIRT3            8200 RFU                 44900 RFU

At page 3 (Materials and Methods), the Authors should expand their descriptions of the procedures followed. For example, how did the authors assess RNA quantity and quality (with the latter being a critical factor for downstream PCR-based reactions)? Furthermore, how was "MDA" assessed? Is a 2.2.4 missing? Please, be advised that MDA is often used as a standards in TBARS-measuring kits.

Response:

RNA quantity and quality were determined by a NanoDrop™ 2000 measuring E260 and ratio E260/E280 with a minimum ratio of 1.8. We included this statement in the methods-section of the manuscript.

MDA was measured by validated stable-isotope dilution gas chromatographic-mass spectrometric (GC-MS) methods. We have briefly included MDA analytics in the methods section and added references. For details, please see the following references:

Tsikas D, Rothmann S, Schneider JY, Suchy MT, Trettin A, Modun D, Stuke N, Maassen N, Frölich JC. Development, validation and biomedical applications of stable-isotope dilution GC-MS and GC-MS/MS techniques for circulating malondialdehyde (MDA) after pentafluorobenzyl bromide derivatization: MDA as a biomarker of oxidative stress and its relation to 15(S)-8-iso-prostaglandin F2α and nitric oxide (NO). J Chromatogr B Analyt Technol Biomed Life Sci. 2016 Apr 15;1019:95-111. doi: 10.1016/j.jchromb.2015.10.009. Epub 2015 Oct 17.

Hanff E, Eisenga MF, Beckmann B, Bakker SJ, Tsikas D. Simultaneous pentafluorobenzyl derivatization and GC-ECNICI-MS measurement of nitrite and malondialdehyde in human urine: Close positive correlation between these disparate oxidative stress biomarkers. J Chromatogr B Analyt Technol Biomed Life Sci. 2017 Feb 1;1043:167-175. doi: 10.1016/j.jchromb.2016.07.027.

At page 3 (Materials and Methods), the Authors state that "EDTA-blood" was used, even though some lines below the 2.2.3 paragraph reports that "serum samples" were processed for subsequent analyses. Please, correct this or elaborate better. Moreover, SIRTs are known to be endocellular enzymes with different and specific localizations. Why do the Authors believe that serum levels of SIRTs may somehow reflect what is found in blood cells or in other tissues (e.g., muscles)?

Response:

The use of blood samples was clarified in the manuscript.

The use of sirtuins in blood was published before (e.g. Tarantino, G.; Finelli, C.; Scopacasa, F.; Pasanisi, F.; Contaldo, F.; Capone, D.; Savastano, S. Circulating levels of sirtuin 4, a potential marker of oxidative metabolism, related to coronary artery disease in obese patients suffering from NAFLD, with normal or slightly increased liver enzymes. Oxid Med Cell Longev 2014, 2014, 920676.).

To our knowledge, there is no evidence that sirtuins are released from solid tissues to the blood. Probably, the sirtuins in the blood reflect sirtuins in blood cells (released after sonication from the intracellular compartment), namely PBMCs. This point is now included in the discussion.

Statistics should also be improved. The Authors state that a One-Way ANOVA-based approach was used, however it is clear that the study design required repeated measures on the same individuals (before and after). Therefore, I strongly suggest to implement the statistics with a factorial ANOVA for repeated measures (pre-post as the repeated factor for analyzing intraindividual differences). That would also provide more robust responses to some gaps in the results and discussion (e.g., page 7, lines 223-224).

Response:

We apologize for an unclear description of statistics. The results of Figure 1, levels of SIRT1, SIRT3 and SIRT5 activity compared pre and post exercise were analysed with a multiple ANOVA with sub groups of pre and post exercise within the three groups. The differences shown are the result of this test. We clarified the statistic description. All further analyses were performed with the difference between pre/post for each sample (Δ enzyme capacity).

At page 4 (Results), a very intriguing result is given (no gender-difference could be found", without investigating further. To me, it requires a little more information or references. Beyond the [41] reference, is there any other literature support that reported convincing evidence of no gender-dependent differences observed in a maximum power output during incremental exercise test on a bycycle ergometer until exhaustion? This looks important, as it might reveal a possible problem with the maximal exercise test performed.

Response:

We checked the data again, corrected the relationship and adjusted the references.

Another thing that captured the reviewer's attention was the table 1. A careful look at the the two columns LOV and VEG revealed that the two groups had not been similarly engaged with the diet regimens. In fact, many more participants had a very long story (>3 yr) with LOV diet, with respect to what seen in VEG group. This could reveal a selection bias. Maybe, the Authors should try to re-analyze data stratifying results for duration of diet. This could provide interesting clues that would be useful for expand discussion.

Response:

We checked the data regarding a difference in duration of diets. We did not find any difference with a two-way ANOVA for the three groups with subgroups of a shorter diet duration (≤1 year) vs longer diet duration (>1 year) for enzyme activities. We chose a cut-off of one year of diet since we think a change of metabolism should have happened after this time period. The tests did not show a difference of SIRT1 (p = 0.7726), SIRT3 (p = 0.6226) and SIRT5 (p = 0.8189) enzyme activities regarding the diet duration. Since there were no differences within the enzymatic activities, we did not re-test further related results like the correlations and did not include these results in the manuscript since there is no additional information using the tests.

In the Results section (page 4), some surprising "significant" differences are reported (e.g., lines 163-165). With similar means and SDs, it would appear that those differences might be due due to chance alone (i.e., null hypothesis true). However, the Authors chose not to show such data in figures as means±SDs. Instead, data were shown as meadian±quartiles. This suggests that these data were not analyzed through parametric approaches. Therefore, in the text means±SDs should not be reported, as it may be misleading.

Response:

We changed all reported data to median (+max/-min) and included a statement in the statistical part of the method section.

Data presented in the figures included basal levels only for SIRT activities. This could obscure other observations that may be of interest. For example, it would be interesting to see the basal levels of SIRT mRNAs, because this could reveal intriguing differences among groups.

Response:

We already tested the basal levels as well as the post exercise levels of sirtuin transcript levels with a multiple ANOVA analysis. There were no significant changes in gene expression levels for SIRT1 (p = 0.7289), SIRT3 (p = 0.7960), SIRT4 (p = 0.8330) and SIRT5 (p = 0.4672). Since there were no further differences in basal levels we only included a note on this in the manuscript (line 227) and only showed the differences pre/post exercise.

With data stratification based on duration of diet regimen, it may be interesting to correlate changes after the acute exercise bout. In fact, it may be observed a differential response of relatively new-LOV (or relatively new-VEG) to the ergometric performance, with respect to the subjects with longer history of those diets. This may provide novel informations to the Authors and the readers.

Response:

We also checked this data regarding a difference in diet duration. Again, we did not find any differences with a two-way ANOVA for the three groups with subgroups of a shorter diet duration (≤1 year) compared with a longer diet duration (>1 year) for ergometric performance given in Watt/kg (p = 0.4246), the change of lactate pre/post exercise (p = 0.5676) and the change of glucose pre/post exercise (p = 0.6337). Since there was no further relation of the ergometric performance with diet duration, we did not include the results in the manuscript.

Some considerations in the discussion seem to be not adequately supported by data. In particular, at page 10, the Authors state that "in vegans, sirtuin capacities decreased or remained unchanged during exercise", and this was explained reasoning about the presumable "decrease in energy deficiency in skeletal/heart during exercise". First of all, the parallel between what is found in the blood and in tissues not that obvious. Secondly, if that were true, how do the Authors explain the fact that the maximal power output and the time to exhaustion was not significantly different when LOV and VEG were compared?

Response:

As stated above there is a correlation of circulating sirtuin levels with different diseases (e.g. Gok O, Karaali Z, Ergen A, Ekmekci SS, Abaci N. Serum sirtuin 1 protein as a potential biomarker for type 2 diabetes: Increased expression of sirtuin 1 and the correlation with microRNAs. J Res Med Sci. 2019 Jun 25;24:56. doi: 10.4103/jrms.JRMS_921_18. eCollection 2019.). The mechanism for circulating sirtuin protein remains unknown so far. Therefore, the metabolic processes can only be speculated.

Retrospectively the authors think that the effect of nutritional diet on maximal power output in this study was concealed by the selection of trained endurance athletes, blood supply to the organs may be a limiting factor, not sirtuins.

We included the issue of differences in sirtuins between blood and solid organs (Discussion-section).

More importantly, MDA was found to increase post-exercise both in LOV and VEG, even though SIRT1 and SIRT3 were observed to respond only in LOV. This may suggest that diet does not control MDA levels post-exercise via SIRT regulation. The Authors did not discuss this at all, and in the reviewer's opinion, this cannot be dismissed sic et simpliciter.

Response:

We added a comment on this in the manuscript’s discussion.

Minor points

As well known, the adaptation of energy supply to exercise is strictly dependent on mitochondrion morphological and functional re-shaping. In this context, the regulation of mitochondrial biogenesis and overall dynamics (fusion, fission, and mitophagy) is critically involved in the exercise-induced metabolic rewiring. The authors almost completely omitted this in their introduction. This should be fixed.  

Response:

We thank the reviewer for this advice. We have revised the introduction.

At page 2 (Introduction), the Authors should expand briefly the part regarding the role of sirtuins in the cellular response to OS, because many key roles of SIRT1 and 3 are omitted (e.g., SIRT3-dependent direct activation of SOD2, SIRT1/FOXO-mediated SODs response to OS etc.)

Response:

We added this information in the introduction.

In the Introduction, the Authors claim that "...literature regarding the influence of exercise performance on sirtuin activity in humans is scarce". A simple and quick search for "exercise sirtuins human" in NCBI returns 50 reviews and 121 items published over the past 10 years. Therefore, I strongly suggest to re-write that statement re-orienting the phrase towards the real novelty of the study, that is the diet-dependent changes.

Response:

Thank you for this comment. Obviously, this statement is not correct and we revised the sentence.

At page 10 (Discussion), the Authors state that "no differences could be observed at the gene expression level before exercise". This was not shown.

Response:

We tested the data at basal levels and did not find any differences. We think it is not necessary to show these data, we might include them as a supplement.

Reviewer 3 Report

In this manuscript “Impact of nutrition on short-term exercise-induced sirtuin regulation: Vegans differ from omnivores and lacto-ovo vegetarians” by Arne Björn Potthast et al., the authors describe a regulation by physical exercise on SIRT1, 3, and 5 according to the diet regimen of the human subject. The manuscript is interesting because connect two aspects that are known to regulate sirtuins activity, i.e.: food intake and exercise. The measurements are conducted on mRNA level and enzymatic activity.

The manuscript is well written and interesting. It is clear, as the authors declares, that blood may represent an underestimation of what is going on in tissues and organs, however, is the best compromise from an ethical point of view. Moreover, it could provide interesting informations.

However, it would be important for the manuscript if authors could address how, according to them and to the literature, sirtuins can be released into the blood. Are they present and measured only in the PBMCs? (this is certainly the case for mRNA they measured). Are they released in microvesicles (exosomes) from tissues due to the stress after physical exercise? (this may be the case for enzymatic assays where serum samples are lysed). In fact, such aspects are not considered in the references (8-11) provided by the authors.

Minor point:

Introduction, line 39: it would be better to write deaceylation instead of deacetylation.

Author Response

Point-by-point response to the comments made by the reviewers

We thank the reviewers for their kind comments and the opportunity to revise our paper. The comments and suggestions have helped to improve the manuscript substantially. In the following, each comment is addressed separately.

Reviewers’ comments:

Reviewer # 3:

The manuscript is well written and interesting. It is clear, as the authors declares, that blood may represent an underestimation of what is going on in tissues and organs, however, is the best compromise from an ethical point of view. Moreover, it could provide interesting informations.

However, it would be important for the manuscript if authors could address how, according to them and to the literature, sirtuins can be released into the blood. Are they present and measured only in the PBMCs? (this is certainly the case for mRNA they measured). Are they released in microvesicles (exosomes) from tissues due to the stress after physical exercise? (this may be the case for enzymatic assays where serum samples are lysed). In fact, such aspects are not considered in the references (8-11) provided by the authors.

Response:

Thank you for this comment. We believe that the sirtuins in blood represent material from circulating cells (after sonication), namely PBMCs, and are not material leaking out of solid organs. To our knowledge, there is no evidence, that sirtuins are transported across the plasma membrane. This has now been included in the Discussion-section. A previous study (Michalak S et al. 2017) reported a correlation between mitochondrial respiration and SIRT3-activity in PBMCs from patients with movement disorders (Reference included).

Minor point:

Introduction, line 39: it would be better to write deaceylation instead of deacetylation

Response:

This has been corrected throughout the manuscript.

Round 2

Reviewer 2 Report

According to the Reviewer's opinion, the major issues of the manuscript were fixed or sufficiently explained.